

# Land cover drives large scale productivity-diversity relationships in Irish vascular plants

Hannah J. White[1], Willson Gaul[1], Dinara Sadykova[2],
Lupe León-Sánchez[2], Paul Caplat[2,3], Mark C. Emmerson[2,3] and
Jon M. Yearsley[1]

[1] School of Biology and Environmental Science, University College Dublin, Dublin, Ireland
[2] School of Biological Sciences, Queen's University Belfast, Belfast, UK
[3] Institute of Global Food Security, Queen's University Belfast, Belfast, UK

Corresponding author
Hannah J. White,
hannah.white@ucd.ie

## ABSTRACT

The impact of productivity on species diversity is often studied at small spatial scales and without taking additional environmental factors into account. Focusing on small spatial scales removes important regional scale effects, such as the role of land cover heterogeneity. Here, we use a regional spatial scale (10 km square) to establish the relationship between productivity and vascular plant species richness across the island of Ireland that takes into account variation in land cover. We used generalized additive mixed effects models to relate species richness, estimated from biological records, to plant productivity. Productivity was quantified by the satellite-derived enhanced vegetation index. The productivity-diversity relationship was fitted for three land cover types: pasture-dominated, heterogeneous, and non-pasture-dominated landscapes. We find that species richness decreases with increasing productivity, especially at higher productivity levels. This decreasing relationship appears to be driven by pasture-dominated areas. The relationship between species richness and heterogeneity in productivity (both spatial and temporal) varies with land cover. Our results suggest that the impact of pasture on species richness extends beyond field level. The effect of human modified landscapes, therefore, is important to consider when investigating classical ecological relationships, particularly at the wider landscape scale.

## INTRODUCTION

Understanding how biodiversity varies with environmental predictors such as energy availability is crucial to improving spatial predictions of how biodiversity will change under scenarios of global change (*Mateo, Mokany & Guisan, 2017*). The relationship between productivity, that is, energy flow through a system, and species richness is a fundamental relationship in ecology that has received much attention, although the idea has often proved to be contentious (*Rosenzweig & Abramsky, 1993*; *Schmid, 2002*) with substantial debate on both the form and underlying mechanisms of the relationship (*Adler et al., 2011*; *Fridley et al., 2012*; *Grace et al., 2012*). Evidence suggests several forms of the productivity-diversity

relationship (increasing, unimodal, and decreasing patterns), often depending on spatial scale (*Waide et al., 1999*; *Mittelbach et al., 2001*). The unimodal form is widely accepted in plant ecology, particularly at the regional scale (*Pärtel, Laanisto & Zobel, 2007*), whilst at larger scales, an increasing monotonic relationship is often observed (*Evans, Warren & Gaston, 2005*; *Gillman & Wright, 2006*), although *Fraser et al. (2015)* demonstrate a unimodal form for plants at both global and regional extents.

A number of potential hypotheses have been proposed to explain the unimodal productivity-diversity relationship frequently observed in plants, with alternative underlying mechanisms influencing the relationship at different spatial scales (*Šímová, Li & Storch, 2013*). These are reviewed in detail in *Rosenzweig (1995)* and *Rajaniemi (2003)*. The majority of these alternative hypotheses consistently rely on the idea of competitive exclusion to explain the initial increase in diversity with productivity, that is, only a small selection of species are able to tolerate low resource levels, therefore as productivity increases, the minimum resource requirements of more species are met (*Rajaniemi, 2003*). A mechanistic explanation for the maintenance of high diversity at intermediate productivities remains elusive, and although competition appears to be the limiting factor at high productivity levels, there is disagreement on why the effect of competition is greater at these levels (*Rajaniemi, 2003*). In contrast, *Michalet et al. (2006)* suggest an alternative role of biotic interactions in the form of facilitation in explaining the rapid increasing phase of the relationship. However, the processes which structure species richness may differ among vs within vegetation types (*Moore & Keddy, 1988*). Biotic interactions of either the competitive or facilitative kind cannot explain the variety of forms of productivity-diversity relationships at large spatial scales, such as the landscape scale, which span multiple vegetation types. This is because biotic interactions alone do not consider regional scale processes such as dispersal (*Zobel & Pärtel, 2008*).

One of the proposed hypotheses for the unimodal productivity-diversity relationship is the environmental heterogeneity hypothesis. Although there have been different formulations of the environmental heterogeneity hypothesis, they all rely on the same basic principle, which is that both very productive and unproductive sites have low resource heterogeneity (*Rosenzweig, 1995*). With increasing productivity, the variety of soil nutrients increases, increasing plant diversity. Past a certain threshold of productivity, however, light becomes a limiting factor (*Tilman & Pacala, 1993*; *Rajaniemi, 2003*), and productivity is spread more evenly over time reducing temporal heterogeneity (*Rosenzweig & Abramsky, 1993*).

*Wright, Currie & Maurer (1993)* present a model for regional species richness under environmental heterogeneity which relies on variation in energy acquisition across species and the amount of available resources across habitats. Environmental heterogeneity is likely most important in shaping the productivity-diversity relationship at regional spatial scales, because other mechanisms (e.g., sampling effects and competitive exclusion) will likely prevail at small spatial scales (*Šímová, Li & Storch, 2013*). The mechanisms supporting this hypothesis, therefore, can be applied at the landscape scale.

An alternative model, the dynamical instability (or equilibrium) hypothesis, suggests that increased productivity reduces the temporal stability of a system as population
dynamics are faster (*Huston, 1979*). Individual populations in highly productive environments are therefore vulnerable to stochastic fluctuations particularly when disturbances are of intermediate frequency (*Rajaniemi, 2003*), leading to increased extinction rates (*Rosenzweig & Abramsky, 1993*).

An outstanding issue is that none of these hypotheses above incorporate the addition of nutrients to a system (*Rajaniemi, 2003*). Enrichment experiments show that whilst nutrient addition increases productivity, the richness of plant communities declines (*Wright, Currie & Maurer, 1993*; *Rajaniemi, 2002*) because a select few species dominate the system (*Clark et al., 2007*). The "paradox of enrichment" (*Rosenzweig, 1971*) suggests that high nutrient load can reduce the stability of a system resulting in local extinction of species and, therefore, a decreased species richness. Highly productive areas are also likely to show homogeneity of nutrient availability; thus, a single dominant species can outcompete other species, reducing the species richness of an area. This is supported by *Stevens & Carson (2002)* who showed that in agricultural fields, the average supply rate of the most limiting resource, rather than resource heterogeneity, explained plant diversity and at high levels of productivity, there will always be one resource in such short supply that species are excluded, and diversity is reduced. Impacts of nutrient addition may be observable at large spatial scales despite acting at smaller spatial scales through wider ecological impacts such as nutrient run off into surrounding habitats (*Swift, Izac & Van Noordwijk, 2004*).

The environmental heterogeneity and dynamic instability hypotheses have all been proposed in reference to relationships established at a regional scale (*Rosenzweig, 1995*). One proposed mechanism underlying the negative phase of the productivity-diversity relationship, which can be applied across ecosystems at the landscape scale, is the species pool hypothesis (*Pärtel, Laanisto & Zobel, 2007*; *Zobel & Pärtel, 2008*). For this hypothesis, the species observed in a particular area are a subset of all species able to tolerate the local conditions. This hypothesis was originally established using macro-evolutionary time scales; however, the idea can also be applied in the context of more recent land use history. The size of the species pool is linked to the historical prevalence of the habitat in question (*Zobel, 1997*; *Zobel & Pärtel, 2008*). Frequently, highly productive areas, such as the improved agricultural pasture prevalent in our study region or the island of Ireland, are more recent additions to the landscape compared to natural grasslands or woodlands. Consequently, there has not been a sufficiently long time for the species pool within these areas to become fully established with all possible species that can tolerate the specific habitat conditions; older, but often less productive, regions are, therefore, more likely to have a higher species richness, suggesting that human activity might impact macroecological relationships. Despite the broad scales at which anthropogenic pressures impact the planet, human influence on macroecological patterns and relationships has been lacking within the literature (*Gaston, 2004*). To address this, recent studies have incorporated land cover and biome type into species-area relationships and showed that agricultural intensity performs as well as biome identity in predicting global species richness, supporting the idea that human factors can augment or even rival environmental factors in explaining macroecological patterns of biodiversity (*Šizling et al., 2016*; *Kehoe et al., 2017*).

The recent availability of high resolution satellite images has increased our ability to estimate productivity of large areas directly using vegetation indices such as the normalized difference vegetation index (NDVI) and the enhanced vegetation index (EVI), as opposed to relying on climate proxies for productivity (*Turner et al., 2003*). NDVI and EVI are commonly used spectral indices which indicate the "greenness" of an area and correlate strongly with net primary productivity and plant biomass (*Box, Holben & Kalb, 1989*; *Evans, Warren & Gaston, 2005*). Regional analyses of the productivity-diversity relationship using satellite-derived productivity data (*Fairbanks & McGwire, 2004*; *Waring et al., 2006*; *Pau, Gillespie & Wolkovich, 2012*) differ from local scale analyses as they integrate multiple land cover types and the signals of productivity and species richness at a regional scale need not have a strong spatial association. For example, a 10 km square containing 60% agricultural pasture will have a strong productivity signal from the pasture but this pasture will likely contribute little to the signal of vascular plant species richness (Irish managed grasslands typically have a species richness less than 20 due to reseeding; L. León-Sánchez, 2017, personal observation). Instead, the signal of vascular plant species richness will be largely due to the 40% of non-agricultural pasture (e.g., hedgerows, woodland). A regional scale analysis can therefore detect how a dominant land cover is associated with the species richness of the surrounding broader landscape. At a local scale the productivity-diversity relationship is known to depend upon land use (*Zhou et al., 2006*), but the impact of the landscape context on the relationship at larger scales remains unknown. This may be particularly relevant to human influences on the landscape, and their impact beyond the scale at which they are implemented.

Using both temporal and spatial measures of EVI (plant productivity) in conjunction with land cover data will allow us to investigate the productivity-diversity relationship at large spatial scales, allowing for processes that operate at scales beyond those of the field scale but also the observable impact of the accumulation of local scale effects. This will let us investigate how human modified landscapes might modulate any established macroecological relationship through landscape scale impacts on biodiversity, through, for example, nutrient leaching (*Swift, Izac & Van Noordwijk, 2004*) or cross-habitat spillover effects (*Tscharntke et al., 2012*). This will allow us to identify the mechanisms underlying the productivity-diversity relationship in vascular plants. As the abiotic environment affects species composition (*Pausas & Austin, 2001*), it is likely that relationships of biodiversity with biotic factors are also impacted. Investigating the productivity-diversity relationship across habitat types, therefore, may contribute to explaining observed variation in relationship form. As the Irish landscape has been heavily modified through agriculture (*Aalen, Whelan & Stout, 2011*), we hypothesize that this impact will have left its signature on spatial patterns of biodiversity beyond the field level, and that including land cover will improve models of the relationship between productivity and diversity.

## MATERIALS AND METHODS
### Data
Species occurrence data for vascular plants were obtained from biodiversity maps from the National Biodiversity Data Centre, Waterford, Ireland (NBDC), which holds biological

records from the Republic of Ireland, and the Centre for Environmental Data Recording, National Museums Northern Ireland (CEDaR) which holds biological records for Northern Ireland. Only records from 1970 onward were included for analyses. Records of sub-species were reclassified to the species level. Only native and archaeophyte species were included in our analyses. Neophytes were excluded on the basis that the mechanisms underlying the productivity-diversity relationship are likely to differ between native and non-native communities (*Korell et al., 2016*). Species status was determined using the Online Atlas of British and Irish Flora (https://www.brc.ac.uk/plantatlas/). Hybrids were also excluded from analyses. This left 1,501,588 records consisting of 998 species for inclusion in these analyses. The majority of observations were recorded at the hectad scale (10 × 10 km). Species occurrence data below this scale was aggregated to the hectad level in order to retain a maximum number of observations for analyses.

MODIS satellite data for the EVI was downloaded at a pixel spatial resolution of 250 m from MODIS composite products from both the Aqua satellite (product MYD13Q1), covering years 2003–2017, and Terra satellite (product MOD13Q1), covering years 2000–2017 (http://modis.gsfc.nasa.gov/). This provides a temporal resolution of 16 days from 2000 to 2002, and 8 days from 2003 to 2017. Data from all time points were used. EVI was used in preference to NDVI because it corrects for aerosol influences and is less affected by saturation when biomass is high, which is particularly relevant for intensive agricultural grasslands (*Huete, 1988*; *Huete et al., 2002*). Negative values of EVI were set to zero as this represents the absence of vegetation within the area. At each time point we aggregated EVI to the hectad scale using the median of the up to 1,600 MODIS pixels within each hectad to match the resolution of the species occurrence data. We also quantified spatial variation in EVI by calculating the standard deviation across all pixels in a hectad. Using the median overcomes issues associated with aggregating erroneous EVI measures at the pixel scale, for example discrepancies due to cloud cover, so that they are unlikely to influence the hectad scale measure of EVI.

Land cover information of 15 land classes at a 25 ha spatial resolution was obtained from level 2 of the CORINE 2012 land cover data (http://www.eea.europa.eu). Dominant land cover in a hectad was taken as the land cover class that had the highest proportion of coverage, if this proportion was greater than 50%. All hectads which consisted of more than 50% water were removed from all further analyses so as not to obscure the results. A hectad was classed as heterogeneous if no land cover had a proportion of coverage greater than 50%. If a hectad was not heterogeneous, then the dominant land cover was taken as the land cover class with the highest proportion of coverage. As pasture (CORINE CLC code 231) dominates the Irish landscape (it is the dominant land cover in 63% of hectads), there were not enough remaining data points to split the additional hectads by dominant land cover. Hectads were therefore labelled as dominated by either non-pasture or pasture, where the CORINE definition of pasture is "Dense grass cover, of floral composition, dominated by graminaceae, not under a rotation system. Mainly for grazing, but the fodder may be harvested mechanically. Includes areas with hedges (bocage)" (*European Environment Agency, 2016*). The three land cover classes of heterogeneous, pasture-dominated and non-pasture dominated were used in

analyses rather than percentage cover of specific classes to avoid multicollinearity with productivity measures.

## Measures of productivity

A series of metrics were used to aggregate the 18-year EVI time series into four measures of productivity within a hectad;

- *Mean*—mean across all time points of a hectad's average EVI (median across pixels). This measure reflects average energy flow within a hectad.
- *Spatial standard deviation*—mean across all time points of a hectad's spatial variation in EVI (standard deviation across pixels). This measure reflects spatial heterogeneity of energy flow within a hectad.
- *Inter-annual temporal standard deviation*—first we calculated the mean in each year of a hectad's average EVI (median across pixels). Then the standard deviation of these within year means was calculated. This measure reflects temporal heterogeneity of energy flow between years.
- *Intra-annual temporal standard deviation*—first we calculated the standard deviation in each year of a hectad's average EVI (median across pixels). Then the mean of these within year standard deviations was calculated. This measure reflects temporal heterogeneity and seasonal variation of energy flow within a year.

The spatial and temporal standard deviations of vegetation measures have previously been used to represent heterogeneity in productivity (*Gould, 2000*; *Levin et al., 2007*). The relationship between these four measures across the island of Ireland are shown in Fig. S1.

## Species richness

Biological recordings data is a rich resource of species occurrences, however, they can be challenging to use for studying spatial patterns of biodiversity due to its often opportunistic collection. Spatial and temporal variation in recorder effort, therefore, is inherent in biological recording data and must be accounted for in its analyses (*Isaac et al., 2014*). We used the program Frescalo to standardise recorder effort between hectads and provide a measure of vascular plant species richness that is comparable among hectads (*Hill, 2012*). Neighborhoods were determined using geographical proximity of the hectad and biological similarity (using Sørensen's similarity coefficient) based on CORINE Land Cover 2012 data (http://www.eea.europa.eu), that is, the 100 most similar hectads from the 200 nearest were classified as the neighborhood. CORINE land cover classes were aggregated into broader classes so that, for example, urban and suburban were not classified as dissimilar to each other as urban and woodland. These classes were: urban; industrial, commercial and transport units; mine, dump and construction sites; artificial, non-agricultural vegetated areas; arable land; permanent crops; pasture; heterogeneous agriculture; forest; shrub and herbaceous vegetation; open ground; inland wetland; coastal wetland; inland water; and marine. We set Φ (the standard neighborhood frequency of species) to 0.8 so that it remained above the 98th percentile of observed values of local

neighborhood frequency for species groups that are not completely recorded. The estimated recorder effort of a focal hectad was then used to scale the "raw" observed species richness for each hectad. Frescalo was carried out using the *Sparta* package (*August, Harrower & Isaac, 2013*) in Rv3.3.3 (*R Development Core Team, 2017*). The Frescalo method has been shown to be a robust method of estimating species richness of both opportunistic data, through comparison with raw occurrence data of moths (*Fox et al., 2014*), and simulated data, where recorder effort and "true" absences were known (*Isaac et al., 2014*).

## Statistical analyses

Frescalo-estimated species richness was modeled as a function of our four EVI measures of productivity using generalized additive mixed effects models (GAMMs) fitted using maximum likelihood. Spatial coordinates (latitude and longitude) were included as an isotropic smooth term (two-dimensional thin plate regression spline), as specified in *Wood (2006)*, to account for structural spatial gradients. Spatial coordinates were also included in the error term using an exponential covariance structure (*Beale et al., 2010*). Including spatial errors accounts for spatial autocorrelation within the model residuals, making Type I error rates more reliable than Ordinary Least Squares and improving model performance compared to models with space only in the covariates (*Beale et al., 2010*).

All covariates except latitude and longitude were fitted using a smooth term with a cubic regression spline. Model selection was implemented using a shrinkage penalty (*Marra & Wood, 2011*). All models were fitted with maximum likelihood and extra null space penalties for smooth terms in the *gamm* function of the R package *mgcv* (*Wood, 2011*). No terms were dropped as all had estimated degrees of freedom greater than or equal to one.

To test hypotheses relating to land cover, a GAMM was applied using the same error structure as in the previous model but separate covariate smooth terms were fitted for hectads dominated by pasture, hectads dominated by non-pasture and heterogeneous hectads. Land cover, as defined by these three classes, was also included as a covariate within the model as recommended by *Wood (2006)*. To further investigate the mechanisms underlying the impact of land cover on the productivity-diversity relationship, a power law species-area relationship between area of the hectad that was not pasture and corrected species richness was fitted using a non-linear least squares model using the R package *nlme*. A final GAMM was applied, similar to the one above, but with an additional smooth of the area of the hectad that was not pasture, thus integrating the species-area relationship into the model. The Akaike Information Criterion of this model was compared with the land cover model above to determine whether including an area parameter improved model performance.

## RESULTS

### EVI relationships

Vascular plant species richness varied spatially across Ireland (Fig. 1) and showed different relationships with different characteristics of EVI (Fig. 2). There was a non-linear

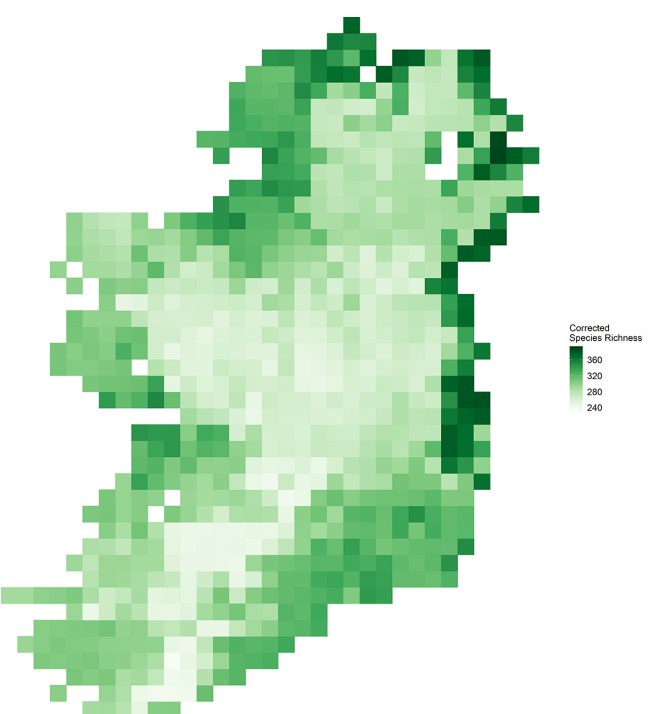

**Figure 1 Corrected vascular plant species richness on the island of Ireland using Frescalo.**

relationship between mean EVI and species richness showing a decrease at higher levels of mean EVI ($F = 14.983$, d.f. = 4.087, $P < 0.001$) whilst species richness increased with the spatial standard deviation of EVI ($F = 8.768$, d.f. = 2.961, $P < 0.001$). Of the two measures of temporal variation in EVI, only the inter-annual standard deviation in EVI showed a relationship with species richness (inter-annual standard deviation; $F = 1.142$, d.f. = 2.309, $P = 0.004$: intra-annual standard deviation; $F = 0$, d.f. = 0.00002, $P = 0.899$).

## EVI and land cover relationships

Incorporating land cover variables into the productivity-diversity model reduced the Akaike Information Criterion from 6,843.558 to 6,817.637, indicating including land cover improves the predictive performance of the model. Species richness showed a steeper downward slope with mean EVI in areas dominated by pasture than heterogeneous areas (Fig. 3). In pasture dominated hectads, species richness decreased with mean EVI ($F = 3.613$, d.f. = 1.521, $P < 0.001$) and increased with spatial standard deviation in EVI ($F = 6.488$, d.f. = 2.275, $P < 0.001$). In heterogenous hectads, species richness also decreased with mean EVI ($F = 1.388$, d.f. = 1.388, $P < 0.001$: Fig. 3) and increased with spatial standard deviation in EVI ($F = 1.977$, d.f. = 1.619, $P < 0.001$: Fig. 4). Hectads dominated by land cover types other than pasture showed no relationship between species richness and mean EVI ($F = 0$, d.f. = 0.0001, $P = 0.237$) or spatial standard deviation in EVI ($F = 0$, d.f. = 0.0004, $P = 0.090$). Species richness showed no relationship with intra-annual standard deviation in EVI in any land cover classification. There was, however, a negative relationship between inter-annual standard deviation in EVI and

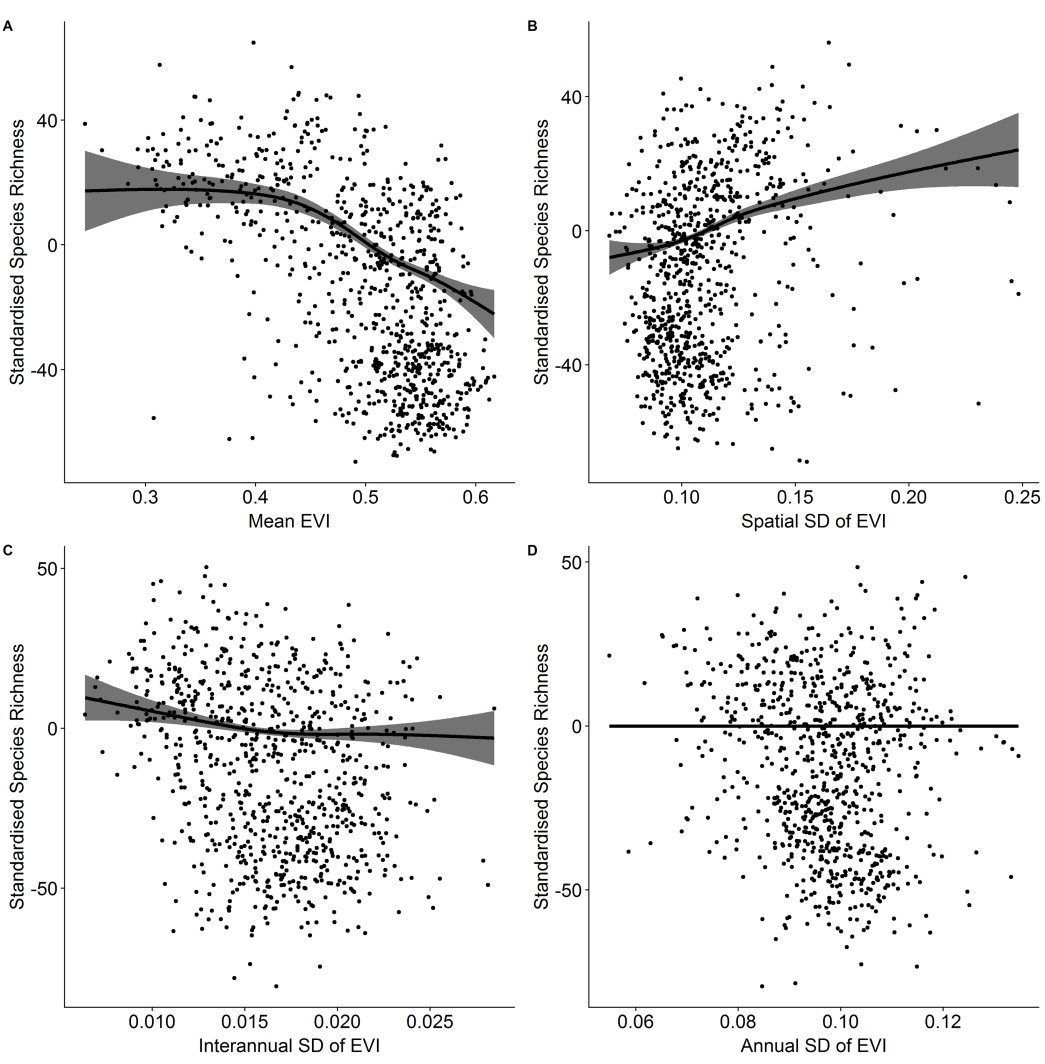

**Figure 2 Standardized species richness as a function of EVI measures.** Standardized species richness of vascular plants estimated using Frescalo in relation to (A) mean EVI, (B) spatial standard deviation of EVI, (C) inter-annual standard deviation of EVI, and (D) intra-annual standard deviation of EVI. The gray areas are the estimated confidence intervals and the points are the model residuals.

species richness in areas dominated by a land cover other than pasture ($F = 0.536$, d.f. = 1.002, $P = 0.016$, Fig. 5). When area of the hectad that was not pasture was included as a smooth within the model, the AIC further reduced to 6,745.789, indicating better predictive performance. In the power law species-area relationship model the scaling coefficient was 0.087 (standard error = 0.004, d.f. = 828, $P < 0.001$; Appendix 1).

## DISCUSSION

The relationship between productivity (mean EVI) and vascular plant species richness in Ireland does not follow the typical hump-back model identified in many previous investigations of productivity-diversity relationships in plants below the continental scale (*Mittelbach et al., 2001*). At high productivity levels, species richness rapidly drops (Fig. 3).

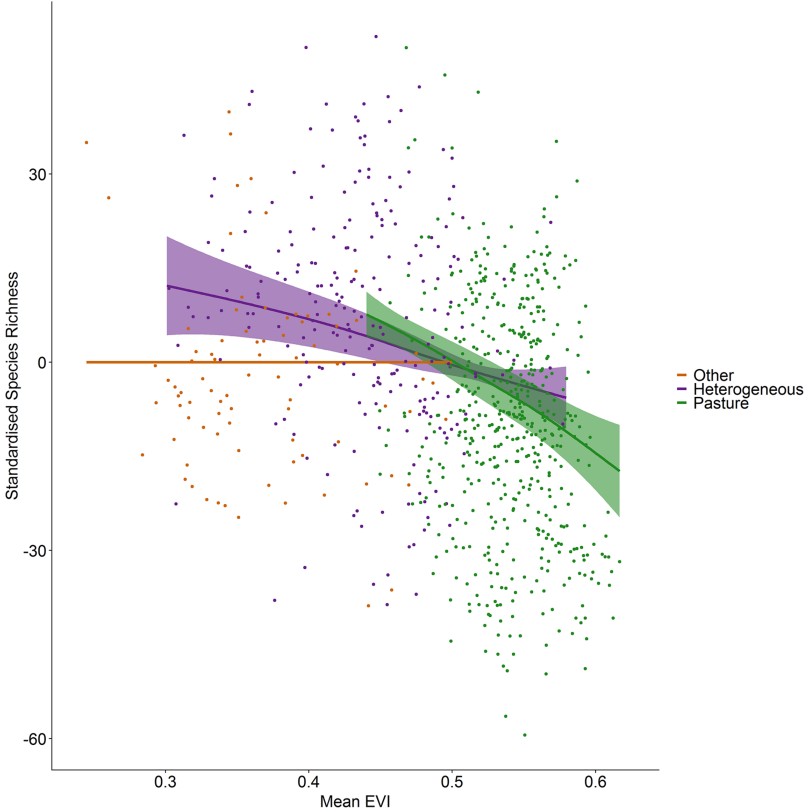

**Figure 3 Standardized species richness as a function of mean EVI smoothed by land cover.** Standardized species richness of vascular plants estimated using Frescalo in relation to mean EVI smoothed by whether the hectad was dominated by pasture, heterogeneous or dominated by a land cover type other than pasture, within a generalized additive mixed effect model. Shaded regions show estimated confidence intervals and the points are the model residuals.               

However, at low productivity levels we find no initial increase in diversity with increasing productivity, providing evidence that the relationships derived from experiments do not always match those found in nature (*Veen, Van Der Putten & Bezemer, 2018*). The highly productive areas in our study were dominated by pasture, as non-pasture dominated areas are primarily inland wetlands (61 of 83 hectads), consisting of inland marshes and peatbogs, as opposed to other productive vegetation classes such as forests and woodlands. The productivity-diversity relationship, therefore, appears to be context-dependent. This supports *Zhou et al. (2006)* who similarly found the productivity-diversity relationship to be strongly affected by different land use types. Human activities which shape the landscape, therefore, can impact broad scale ecological relationships.

The island of Ireland has undergone a large transformation in land use over the last 500 years (*Aalen, Whelan & Stout, 2011*), with the large deforestation and conversion to agricultural land in the 16th–18th centuries (*Aalen, Whelan & Stout, 2011*; *Everett, 2015*). This transformation is reflected in the CORINE data, with only one hectad having more than 50% land cover of class *forest*, and most of the landscape being dominated by pasture (*Hall, 1997*). However, agricultural pasture in Ireland typically has a low vascular

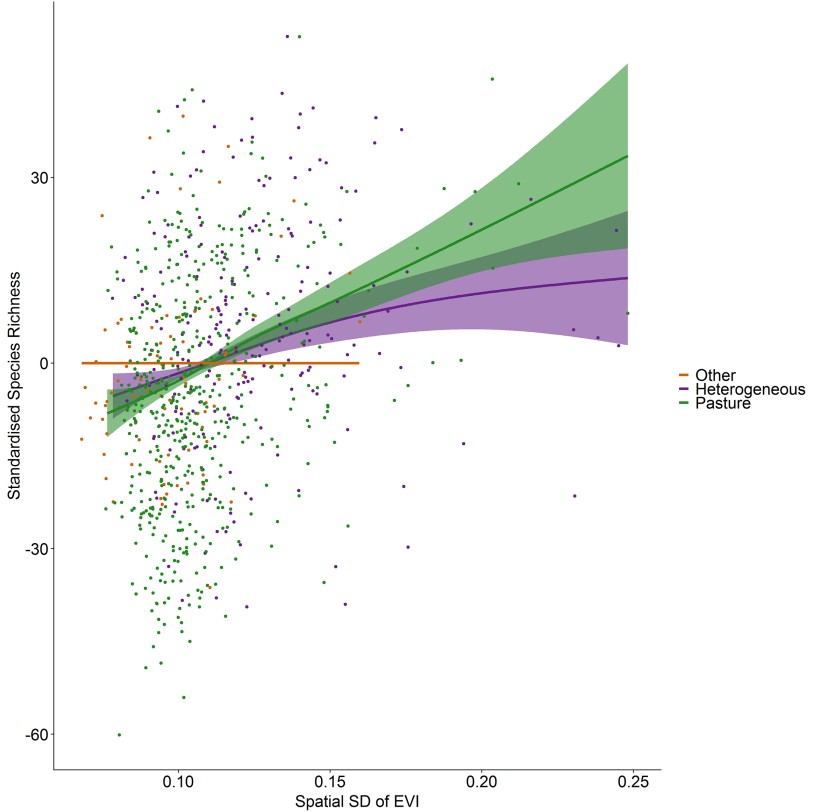

**Figure 4 Standardized species richness as a function of the spatial standard deviation in EVI smoothed by land cover.** Standardized species richness of vascular plants estimated using Frescalo in relation to the spatial standard deviation in EVI smoothed by whether the hectad was dominated by pasture, heterogeneous or dominated by a land cover type other than pasture, within a generalized additive mixed effect model. Shaded regions show estimated confidence intervals and the points are the model residuals.

plant species richness as a result of management practices aimed to increase productivity including high levels of nitrogen addition (*Pallett, Pescott & Schäfer, 2016*), and sowing and reseeding with a limited number of species. In Ireland, these are perennial ryegrass *Lolium perenne*, Italian ryegrass *L. multiflorum,* and white clover *Trifolium repens* (*Department of Agriculture, Food and Marine, 2017*). Despite the potential to sow high diversity seed mixes, natural colonization is often suppressed in sown communities, leading to both taxonomic and functional homogenization of plant communities (*Veen, Van Der Putten & Bezemer, 2018*). In pasture-dominated areas, the biodiversity signal we calculated is coming from the fragmented parts of natural land cover types and field margins, rather than the fields themselves which only exhibit a handful of species (L. León-Sánchez, 2017, personal observation), particularly when it is taken into consideration that only native species were included in our analyses. The active influence of humans on the landscape where productive land covers such forests have been cut down, and only remnant fragments exist (*Aalen, Whelan & Stout, 2011*; *Everett, 2015*), has left the most productive land cover type on the island of Ireland to be pasture, demonstrating the impact of human activity on productivity-diversity relationships at the landscape scale.

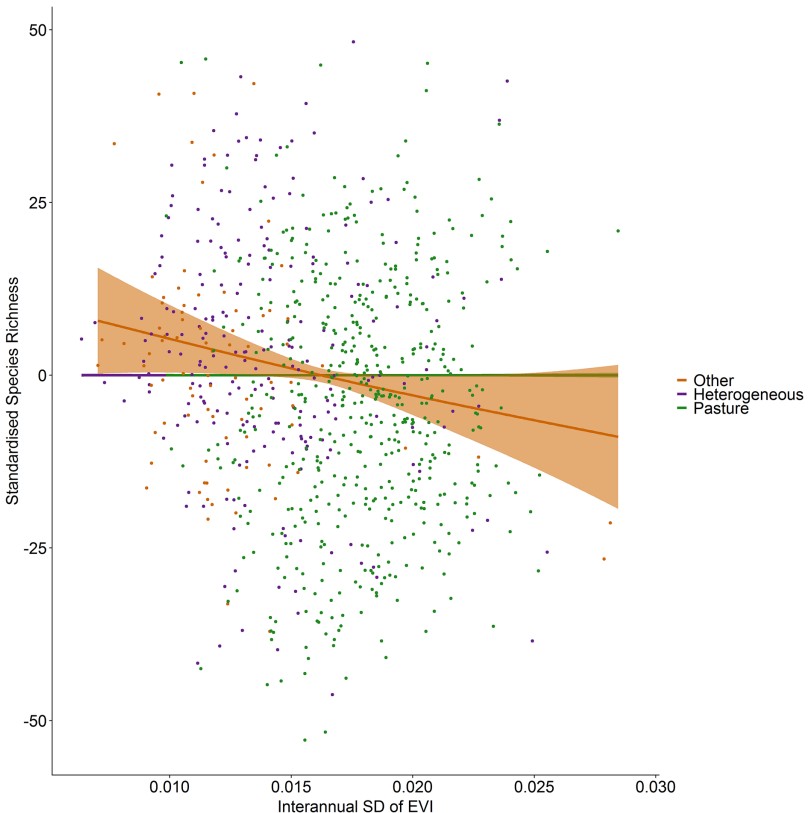

**Figure 5 Standardized species richness as a function of inter-annual standard deviation in EVI smoothed by land cover.** Standardized species richness of vascular plants estimated using Frescalo in relation to the inter-annual standard deviation in EVI smoothed by whether the hectad was dominated by pasture, heterogeneous or dominated by a land cover type other than pasture, within a generalized additive mixed effect model. Shaded regions show estimated confidence intervals and the points are the model residuals.

A species-area relationship is one possible explanation of our productivity-diversity relationships (*Rosenzweig, 1995*). Hectads with a high proportion of agricultural pasture (and therefore high productivity) have a lower area of "natural" habitat (e.g., field margins and wooded areas), but the biodiversity signal is predominantly coming from these "natural" habitats. Therefore, the area available for most vascular plant species within a hectad is closer to the area of non-pasture land-cover, rather than the entire area of a hectad. We do find a positive relationship between area of non-pasture in a hectad and vascular plant species richness (see Supplemental Information). Including the area of non-pasture as an additional smooth term in our model for productivity-diversity relationships also results in improved model performance and fully explains the negative relationship of richness with mean EVI in heterogeneous hectads. However, in pasture-dominated hectads the negative relationship between mean EVI and species richness is maintained, even after correcting for area of non-pasture. This indicates that reduced area of suitable habitat is not sufficient to fully explain the decrease in species richness with increasing pasture productivity. The negative productivity-diversity relationship in hectads dominated by pasture, therefore, is more than a species-area

relationship, and the association between productive pasture and species richness extends beyond the field-scale to a landscape scale.

Several hypotheses have been proposed to explain decreases in species richness at high productivities, with competitive exclusion frequently underlying them (*Rosenzweig, 1995*; *Stevens & Carson, 2002*; *Rajaniemi, 2003*). For example, in agricultural fields, such as those that dominate the Irish landscape, *Stevens & Carson (2002)* suggest that one resource will always be a limiting factor. In productive systems in particular, the interaction between root and shoot competition can indirectly structure communities, thus affecting species richness (*Lamb & Cahill, 2008*). The roles of evolutionary history (*Pärtel, Laanisto & Zobel, 2007*; *Zobel & Pärtel, 2008*), dispersal (*Pärtel & Zobel, 2007*), and facilitation (*Michalet et al., 2006*) within the productivity-diversity relationship have also been proposed. However, highly agricultural landscapes in temperate regions, such as those on the island of Ireland, have likely existed for too short a period to allow for evolution or historical migration of species to increase the species pool (*Pärtel, Laanisto & Zobel, 2007*).

Facilitation may explain our observed lack of a positive relationship between productivity and diversity at low productivities. *Michalet et al. (2006)* suggest that facilitation as well as competition can drive the productivity-diversity relationship, particularly when considering disturbance, when the realized niche of stress-intolerant species is increased in severe conditions. If we assume a high level of disturbance in areas with low productivity (low mean EVI), then the absence of an increasing phase in species richness may suggest facilitation between species in this area as we observe higher species richness than is expected from the classical hump-shaped model.

The predictive power of the productivity-diversity model in the present study was greatly improved when land cover was incorporated into the model. Moreover, the productivity-diversity relationships varied between pasture-dominated, non-pasture dominated, and heterogeneous land cover classes. This supports previous suggestions that integrative models have higher explanatory power than the traditional bivariate models of productivity and diversity (*Grace et al., 2014*, *2016*). The relationships found when the landscape was dominated by pasture differed substantially to those across the entire study, and in particular were responsible for driving the steep downward phase of the relationship at high levels of mean EVI, similar to the finding of a global of plant species richness which found that the hump-shaped productivity-diversity relationship changed to a positive linear effect when sites of anthropogenic origin were removed (*Adler et al., 2011*). *Moore & Keddy (1988)* suggest that different processes structure species richness patterns between vegetation types. However, the differences between pasture-dominated hectads and the other two land-cover classes may also reflect the relative decreased variation in productivity within pasture, rather than different processes (*Mittelbach et al., 2001*). Notably, our results contrast with those of grazing lands in semi-arid Mediterranean rangelands where in grazed plots, species richness increased along a productivity gradient (*Osem, Perevolotsky & Kigel, 2002*). In grazed pastures in Ireland, however, there is a strong decline in species richness with productivity.

Fertilization and grazing, crucial characteristics of pasture, have been shown to have both direct and indirect negative effects on species richness in grasslands (*Socher et al., 2012*;

*Isbell et al., 2013*), yet none of the proposed hypotheses for the downward phase of the hump-shaped productivity-diversity relationship, take into account nutrient addition (*Rajaniemi, 2003*). Supplementary fertilizer, either organic or inorganic, is frequently applied to pastures to increase yield for fodder. This increased yield, however, appears to be to the detriment of diversity and corresponds with the "paradox of enrichment" in that additional nutrient load destabilizes the steady state of a system (*Rosenzweig, 1971*). The contribution of pasture-dominated sites to the productivity-diversity relationship in Ireland suggests land management practices such as nutrient enrichment and reseeding are a vital component of spatial variation in species richness and their impacts reach beyond the scale at which they are applied to the broader landscape. This raises the question of how broad, in fact, are the ecosystem effects of these activities? For example, how great is the impact of leaching from nitrogen addition to surrounding biodiversity? Further, the lack of an increasing phase in species richness with mean EVI, and no relationship between the two in hectads dominated by land cover types other than pasture supports the suggestion that mature plant communities arising from natural assembly processes favor high biodiversity and show no relationship between species richness and ecosystem functioning variables such as primary productivity (*Roscher et al., 2016*).

The positive relationships between spatial heterogeneity in EVI and species richness, not only across the entire dataset, but particularly in pasture-dominated hectads also supports the species-area hypothesis highlighted above. Hectads that show higher spatial heterogeneity in productivity are likely to be those that, despite being dominated by pasture, are integrated by other land covers including larger proportions of scrubland, wooded areas and field margins. Despite low biodiversity in pasture habitat, therefore, the surrounding landscape can still host a large number of species. This supports the idea that to maintain biodiversity in agricultural areas, the overall landscape mosaic should be heterogeneous and structurally complex (*Tscharntke et al., 2005*), and that fragmented landscapes can still support a high level of biodiversity as long there is not substantial habitat loss to push biodiversity down the species-area curve (*Fahrig, 2013*). Incorporating both local and landscape level vegetation index measures can substantially improve the explanatory power of models of plant species richness (*Parviainen, Luoto & Heikkinen, 2009*), emphasizing the role of landscape effects on local processes. Further, in areas showing heterogeneity in land cover, increasing spatial heterogeneity in EVI increases species richness, supporting the environmental heterogeneity hypothesis (*Diamond, 1988*).

In contrast with spatial heterogeneity in EVI, species richness showed a slight decrease with inter-annual standard deviation of EVI, although the effect is consistent with the relationship found by *Levin et al. (2007)* for perennial plants. Further, there was no relationship with intra-annual EVI, and although *Fairbanks & McGwire (2004)* showed that the significance of intra-annual NDVI to species richness varied between habitats in California vegetation types, none of the smooths by land cover type for intra-annual variation in EVI were significant at the 5% level in our study. In fact, temporal heterogeneity of productivity appeared generally unimportant to vascular plant richness in our study as the only significant relationship found was the negative relationship of the inter-annual standard deviation of EVI in areas dominated by land covers other than pasture.

Many biodiversity relationships are scale-dependent (*Willis & Whittaker, 2002*), including the productivity-diversity relationship in plants (*Waide et al., 1999*; *Mittelbach et al., 2001*; *Gillman & Wright, 2006*). The prevalence of different shapes of the relationship within the literature varies between local, landscape, regional, and global scales, for example, unimodal relationships are found more often at fine-scale resolutions than more coarse-scale resolutions (*Gillman & Wright, 2006*). Therefore, different mechanisms and processes are likely to interact with and influence the relationship at different spatial scales. Our study provides a large-scale evaluation of the relationship taking into account the heterogeneity in land cover that exists at this scale. Although human influence on species richness has often been acknowledged at a local spatial resolution, human activities also impact large scale spatial patterns of diversity including the species-area relationship (*Kehoe et al., 2017*) and the elevational richness gradients (*Nogués-Bravo et al., 2008*). Through linking classical ecological theory with landscape ecology, our results provide further evidence that at large scales, human dominated landscapes can impact macroecological models of biodiversity partially through limiting the area of suitable habitat for species to persist. The large proportion of pasture across the island of Ireland provides a useful, large-scale opportunity with which to investigate this impact as high levels of nutrient addition and reseeding of limited species greatly alter individual growth, competitive exclusion and, subsequently, community assembly.

## CONCLUSIONS

The use of satellite data is becoming increasingly important for habitat mapping, modeling of species distributions and predicting the impacts of environmental change on ecosystems. Here, the availability of data describing broad scale vegetation indices has allowed a regional assessment of the productivity-diversity relationship. When applied in the context of ongoing land cover change and increasing land use intensity, the results show that human impacts can alter classical ecological relationships at broad spatial scales, and that our efforts to maximize productivity might have detrimental consequences for biodiversity across the landscape.

## ACKNOWLEDGEMENTS

We thank the many citizen scientists who collected and contributed their data to the National Biodiversity Data Centre (NBDC) and the Centre for Environmental Data and Recording. We thank Tomás Murray for access to the NBDC and data and his comments on an earlier version of this manuscript. This work was carried out as part of the Grassland Resilience Project.

### Funding

This publication has emanated from research conducted with the financial support of Science Foundation Ireland and the Department for the Economy, Northern Ireland under

Grant number (15/IA/2881). The funders had no role in study design, data collection and analysis, decision to publish, or preparation of the manuscript.

## Grant Disclosures
The following grant information was disclosed by the authors:
Science Foundation Ireland.
Department for the Economy, Northern Ireland: 15/IA/2881.

## Competing Interests
The authors declare that they have no competing interests.

## Author Contributions
- Hannah J. White conceived and designed the experiments, analyzed the data, contributed reagents/materials/analysis tools, prepared figures and/or tables, authored or reviewed drafts of the paper, approved the final draft.
- Willson Gaul contributed reagents/materials/analysis tools, authored or reviewed drafts of the paper, approved the final draft.
- Dinara Sadykova authored or reviewed drafts of the paper, approved the final draft.
- Lupe León-Sánchez authored or reviewed drafts of the paper, approved the final draft.
- Paul Caplat authored or reviewed drafts of the paper, approved the final draft.
- Mark C. Emmerson authored or reviewed drafts of the paper.
- Jon M. Yearsley conceived and designed the experiments, authored or reviewed drafts of the paper, approved the final draft.

## Data Availability
EVI data is available for download from http://modis.gsfc.nasa.gov/. Vascular plant distribution data for the Republic of Ireland is available on request from the National Biodiversity Data Centre (http://www.biodiversityireland.ie/projects/vascular-plants/database/; http://www.biodiversityireland.ie/contact-us/), and for Northern Ireland from the Centre for Environmental Data and Recording (https://www.nmni.com/CEDaR/CEDaR-Information-service.aspx). CORINE Land Cover data is available for download from http://www.eea.europa.eu. Code for the analyses are available at https://github.com/HannahWhite/Prod_div.

## Supplemental Information
Supplemental information for this article can be found online at http://dx.doi.org/10.7717/peerj.7035#supplemental-information.

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
