# Peer review of "Land cover drives large scale productivity-diversity relationships in Irish vascular plants"

_PeerJ, doi:10.7717/peerj.7035_

## Round 0.1 · original submission · Major Revisions

I have received two reviews, one is positive and the other one is rather critical. I concur with most of the comments and urge you to comply with the comments.

·

Basic reporting

This article meets the standards in every way.

The text is clearly and professionally written, with excellent organization. The paper as a whole is easy to follow and to understand.

The introduction provides a comprehensive review of patterns and hypotheses linking productivity and diversity. This section is well-organized and thorough. My initial thought was that references to the Nutrient Network experiments (Grace, Adler) would be appropriate, but I see that they are included in the discussion. Otherwise, the relevant literature is cited, and it provides a context for the current study.

The paper is structured with appropriate sections. The figures are relevant and are labelled and described well.

The raw data in this study came from publicly-available sources, rather than being generated by the authors. The original sources of the data are provided and the analysis code is available.

The results are relevant to the questions raised in the introduction, and the paper is a self-contained unit.

Experimental design

This article meets the standards in every way.

This paper begins by reviewing what is known about patterns linking productivity and biodiversity of plants, and reviewing the proposed mechanisms that create those links. Although it is no longer the “hot topic” it used to be, the question of how productivity influences diversity remains unsettled. The authors identify a few important gaps that their study can address. First, the effect of anthropogenic nutrient additions may differ from the effect of naturally high productivity. Second, much of the work on this question has been done at very large or fairly small scales, whereas a regional scale allows for better understanding of the role of heterogeneity in land use. This study covers a regional area at a large grain size (10 km square units), and includes a large proportion of pasture land, so it is well suited to address these two gaps.

The methods for quantifying land use and productivity appear to be logical and sound. The data analysis methods are complex and unfamiliar to me (and probably to many readers) but are described clearly and justified well. I would like to see a little more description of the plant data. What are the records that were used – species lists for certain areas, maps of species distributions?

Validity of the findings

This article meets the standards in every way.

The authors are able to draw sound conclusions from their analysis. By comparing pasture, non-pasture, and heterogeneous land cover, they can address questions about effects of human impacts and land use. The use of various measures of spatial and temporal heterogeneity in productivity makes it possible to draw some conclusions about mechanisms causing the observed patterns. Those conclusions are limited by the available data, and the authors don’t overstep those limits.

Additional comments

This is a very nice contribution to the field!

·

Basic reporting

White and others report the results of a really interesting study. Their approach of using 10x10km hectads covering the whole island of Ireland is certainly novel for studying productivity-diversity relationship in plants. The idea is good and the results are interesting. However the method section has some gaps, which do not allow to assees the validity of the analysis. And there are also some issues with the theoretical aspects of the study that might need a bit more work. Overall, though, the manuscript is very promising.

The manuscript is written in very good English. The text is clear and unambiguous. Confusing passages are not obscure because of the language, but because of the content that is not fully described, etc.

It is difficult to give a thorough overview of all the studies related to vegetation productivity-diversity relationship (PDR). And it is impossible to do it within one journal paper. There is just not enough space. So the authors have to come up with an angle that covers the topic from the aspects that they consider the most novel and interesting in their analysis. White et al focus on mechanisms behind PDR, like heterogeneity, fertilization, ecosystem stability, historical and evolutionary background. However, most of the mechanisms discussed are operating within ecosystems, and it remains unclear from the Introduction how, for example, ecosystem stability or fertilizing work on the hectad scale. Also, in the literature, most studies that have found positive relationship between heterogeneity and diversity have been conducted within an ecosystem, and not between them on such an understudied scale as 10x10 km grids. The Introduction does not really exaplain how the described mechanisms could work on hectad scale, which a) has a different spatial resolution, and b) cuts different ecosystems into random fragments and puts them together in artificial grid squares. I do not want to say that it´s either bad or wrong, but I´d like to see at least some attempts to try to translate the mechanisms on this novel spactial scale.

This translation/transition has been attempted with historical and evolutionary history mechanism (lines 96-109), but it is not very convincing, because while Pärtel et al 2007 analysis dealt with biogeographical-scale processes, claiming that the PDR in temperate zone habitats, especially grasslands are affected significantly by the last Ice Age (which obliterated high productive habitats from temperate zones, while tropical habitats have enjoyed relatively stabile conditions for the past couple of million of years), White and others assume or expect the same kind of outcome from human activity ("Frequently, highly productive areas such as farmland are more recent additions to the landscape compared to natural grasslands or woodlands. Consequently, there has not been a sufficiently long evolutionary time for the species within these areas to undergo significant speciation; older, but often less productive, regions are, therefore, more likely to have a higher species richness, suggesting that human activity might impact macroecological relationships."). So, assumedly the authors try to replace the effect of the Ice Age with human influence on habitats, because Ireland is largely covered by pastures, which are artificial (? - more about that later) habitats, and therefore the negative PDR is expected mainly because of human influence, but not because of geological history. Just that both processes kind of have the same outcome. It´s and interesting idea, but based on the analysis and data in this study, it is impossible to separate these effects as Ireland has both a lot of pastures, but at the same time it was also covered with the permanent ice during the last glacial maximum. In order to understand the role of evolutionary processes, like spaciation, in generating habitat-specific diversity patterns, it is necessary to include some kind of assessment of past habitats and land cover (eg Zobel et al 2011 in GEB for Canary islands). There seem to be studies that have mapped the potential natural habitat cover for Ireland (e.g. Cross 2006 in Biology and Environment: Proceedings of the Royal Irish Academy "The potential natural vegetation of Ireland"), and by dividing species pool according to habitat preferences, it would be possible to separate these effects. But not based on the analysis presented in this manuscript.

So, in the current form, the Introduction is not very well integrated with the rest of the manuscript, and is a bit misleading about the constraints of the potential interpretation of the analysis.

Experimental design

The issue of promising a bit too much comes up again when the authors describe the objectives of the study. The main research goal is expressed in lines 131-4: "Using both temporal and spatial measures of EVI (plant productivity) in conjunction with land cover data will allow us to investigate the productivity-diversity relationship at large spatial scales, and how human modified landscapes might modulate any established macroecological relationship through landscape scale impacts on biodiversity."; the authors add later (L 139-42), that: "As the Irish landscape has been heavily modified through agriculture (Aalen, Whelan and Stout, 2011), we hypothesise that this impact will have left its signature on spatial patterns of biodiversity beyond the field level, and that including land cover will improve models of the relationship between productivity and diversity." Additionally, I will deal with the "temperate and spatial measures of EVI" below.

Now, the Methods section. There are couple of issues that needs to be pointed out.

1) Vascular plant data or Species occurrence data (lines 145-55). The authors do not describe the data at all. Is it occurrence data like in GBIF? Or vegetation survey´s data? Or some combination of different data types? What are the sources of this data - NBDC has not probably recorded all the data by themselves? How many species there were in the dataset? How many records? What was the location precision threshold? Very important is also the timeframe of data. Assumedly only records from certain time period were used? Otherwise the diversity data would not fit with the land use data. In lines 290-1 the authors say that "The island of Ireland has undergone a large transformation in land use from its natural state (Aalen et al. 1997), with most of the landscape dominated by pasture.", but I could not find a timeframe during which this transformantion happened.

In lines 153-4 the authors say that "Species occurrence data was aggregated to the hectad level (10 km square) in order to retain a maximum number of observations for analyses." First of all - a hectad should be 100 km square, not 10 km? But secondly - Ireland has several vegetation maps dating from different time periods. The authors also cite this: Online Atlas of British and Irish Flora. And this atlas has been using the same 10x10 km hectad scale. My first assumption, when agreeing to review this manuscript, was that the authors have used vegetation atlas data for this PDR. Because the spatial scale is exactly the same, and the vegetation survey´s that are the basis of vegetation atlases have been done really systematically and by using the same methodology in all the hectads. Plus, the data is freely available. But after going through the "Data" section it seems that the authors have not used the vegetation atlas data, but instead something else. Why?

2) Irish landscapes. For a person not very familiar with the nature in Ireland, it remains realtively vague, what exactly is a pasture. The passage that provides some description is in the Discussion (lines 290-303). Something like that should be in the Introduction or at least in Methods. And it should give a bit more precise overview of that particular habitat type. For example, in lines 290-1 it says that "The island of Ireland has undergone a large transformation in land use from its natural state (Aalen et al. 1997), with most of the landscape dominated by pasture." the authors do not provide the timeframe of this transformation. Nor the range of it. What were the habitats that were replaced by pastures? When did the massive fertilization with the nitrogen started? Or the sowing and reseeding? Are these processes still going on, or have they intensified? As the authors core idea is to find signs of human influence on PDR on macroecological scale, knowing the duration of these processes is crucial information.

3) Hectad grouping. The analyzed hectads are divided into three groups, based on the land use dominance (hectad is 50% or more covered by one specific of habitat or type of land use):pasture-dominated, heterogeneous, non-pasture dominated. More than half of the hectads are pasture-dominated, while there seems to be altogether two hectads dominated by urban land use; zero hectads by industrial, quarry and artificial vegetation; 4 by arable land; 6 by heterogenous agriculture (btw, where these hectads classified as heterogeneous or non-pasture dominated?); one by forest; 9 by scrubs; 61 by inland wetlands; and zero hectads where both inland and marine waters were dominating. So, 61 hectads dominated by inland wetlands vs 22 that are dominated by all other habitats and land use types.

The long list extracted from the GitHub repository data matrix is meant to show what a strange construction this non-pasture dominated hectad group is. It feels like more adequate title for it would be "inland wetlands" (after removing the few non-wetland hectads). That would make things much clearer. Because when you look at figure 3, it´s striking how pasture hectads are the most productive, then come the heterogeneous ones, and the non-pasture dominated hectads are the least productive. One would expect that both heterogeneous and non-pasture dominated terrestrial habitats would be on average more productive than species poor grasslands, because other habitat types would typically contain woody species, which have significantly more aboveground biomass than pasture vegetation, which has more or less only herbacous layer.

Ireland´s land cover is what it is, and I don´t want to blame authors for not having more forest hectads etc in the dataset. I just wanted to show how the explanation of why pastures are the most productive hectad type in this data is buried too deep. That information of the non-pasture dominated hectads comprise mainly of wetlands should be included in the main text (or as a table or sth). The fact that forests and woodlands seem to be cut down and the remnants are very fragmented, which has generated a situation where the most productive land use type is pasture, which is heavily influenced by human activities, is in my mind a very convincing argument supporting what the authors are claiming - that human influence affects PDR in macroecological scale.

4) Hectad filtering. Going back to the same GitHub repository data matrix I was surprised to see how there were no hectads dominated by waters. Because of some big lakes (e.g. Lough Neagh that covers nearly 400 sq km) and very rugged coastline I was expecting to find hectads dominated by waters and only including a tip of a peninsula etc. But no. Did you remove any of the hectads from the dataset? The island area is 84421 sq km, but there are only 830 square shaped hectads in the dataset. Something seems to be missing, but the Methods section does not mention anything about it.

5) EVI calculations. MODIS satellite data covers the years 2000-2017 (2003-2017 for aquatic systems), and if I understood correctly, you used the data available within this timeframe. Then you scaled the 250 m pixel scale into hectad scale, but instead of calculating the total EVI for the hectad, you used median value (lines 162-4). Why? Why not calculate the total EVI for each hectad in each time point? Or at least have the mean value - that should contain less noise in representing the productivity of an hectad. Using median means that it´s impossible to know which land use type represents the productivity value in heterogeneous hectads, which in turn makes interpreting the results much more difficult.

Another question regarding EVI calculations is the "time point". You never define in the text, what is one time point in your study. Are these daily measurements like MODIS takes, or mean values for some time period? Did you use EVI data for the whole year, or only for vegetation period? This last question is also significant when it comes to comparing the results of this study with the other PDR studies, because the latter ones almost always use peak aboveground biomass as the proxy for productivity. Using data for all seasons and for different land use types dominated by different vegetation (e.g. evergreen vs deciduous/herbaceous vegetation) could have "side-effects" to the productivity estimations that are difficult to predict. So, the process of how staellite data was transformed into EVI values per one hectad needs to be described in more detail.

The matter of how EVI was calculated affects also the measures of productivity. In lines 179-197 the authors describe 4 productivity measures that they used for the analysis. First of all, are the measurements correlated with each other? But more importantly - how can the authors be sure that the second measure "Spatial standard deviation" (lines 184-6) indeed measures spatial variation of EVI? Because they define the measure as: "mean across all time points of a hectad’s spatial variation in EVI (standard deviation across pixels).". But like I described in the previous paragraph, the variability of productivity in one hectad could also be the result of seasonal variation, and differences in seasonal productivity dynamics between different land use types. And as the hectad´s productivity is calculated based on the median EVI value on all the 1600 pixels in each hectad - does that mean that the basis of productivity variability in each hectad could be calculated based on different pixel values? Because the median-valued pixel is probably different in different time points. That could really make the producitivty variability measurements just random numbers.

Validity of the findings

As I´ve had quite a few critical things to point out for the Methods, I will not focus too deeply in the Discussion section in this round of reviews. I just want to point out one thing. The Discussion starts with summarizing the main results: "The relationship between productivity (mean EVI) and vascular plant species richness in Ireland does not follow the typical hump-back model identified in many previous investigations of productivity-diversity relationships in plants below the continental scale (Mittelbach et al. 2001)." (lines 279-281). Are hump-back and negative PDRs really a different thing? If we would sample the entire productivity gradient, negative PDR would be impossible - if there is 0 species there is also 0 productivity and vice versa. So, from that point of view it is very difficult to oppose hump-back PDR with negative PDR. Statistically these relationships are different, but from the ecological point of view not so much. (See also Pärtel et al 2010 Ecology "The productivity–diversity relationship: varying aims and approaches" for more discussion about that.) I would be more careful in building up the excitement in the Discussion by confronting these results.

Additional comments

In conclusion - the study is based on a very interesting idea. Methods section has some gaps at the moment that obstruct comprehensive assessment of the scientific outcome of this study.

---

## Round 0.2 · accepted · Accept

I received two sets of positive reviews from the reviewers. Therefore, your submission has been recommended for publication in PeerJ.

·

Basic reporting

no comment

Experimental design

no comment

Validity of the findings

no comment

Additional comments

The improved version of the manuscript is very good. All the theoretical gaps pointed out in the review are now filled, also the methods are now sufficiently explained. It´s a very interesting study and I would recommend to publish it.